# Machine Learning and Virtual Reality on Body Movements’ Behaviors to Classify Children with Autism Spectrum Disorder

**DOI:** 10.3390/jcm9051260

**Published:** 2020-04-26

**Authors:** Mariano Alcañiz Raya, Javier Marín-Morales, Maria Eleonora Minissi, Gonzalo Teruel Garcia, Luis Abad, Irene Alice Chicchi Giglioli

**Affiliations:** 1Instituto de Investigación e Innovación en Bioingeniería (i3B), Universitat Politécnica de Valencia, 46022 Valencia, Spain; malcaniz@i3b.upv.es (M.A.R.); jamarmo@i3b.upv.es (J.M.-M.); meminiss@upvnet.upv.es (M.E.M.); gonzaloteruelg@gmail.com (G.T.G.); 2Red Cenit, Centros de Desarrollo Cognitivo, 46020 Valencia, Spain; lam@redcenit.com

**Keywords:** autism spectrum disorder, body movements, repetitive behaviors, virtual reality, machine learning

## Abstract

Autism spectrum disorder (ASD) is mostly diagnosed according to behavioral symptoms in sensory, social, and motor domains. Improper motor functioning, during diagnosis, involves the qualitative evaluation of stereotyped and repetitive behaviors, while quantitative methods that classify body movements’ frequencies of children with ASD are less addressed. Recent advances in neuroscience, technology, and data analysis techniques are improving the quantitative and ecological validity methods to measure specific functioning in ASD children. On one side, cutting-edge technologies, such as cameras, sensors, and virtual reality can accurately detect and classify behavioral biomarkers, as body movements in real-life simulations. On the other, machine-learning techniques are showing the potential for identifying and classifying patients’ subgroups. Starting from these premises, three real-simulated imitation tasks have been implemented in a virtual reality system whose aim is to investigate if machine-learning methods on movement features and frequency could be useful in discriminating ASD children from children with typical neurodevelopment. In this experiment, 24 children with ASD and 25 children with typical neurodevelopment participated in a multimodal virtual reality experience, and changes in their body movements were tracked by a depth sensor camera during the presentation of visual, auditive, and olfactive stimuli. The main results showed that ASD children presented larger body movements than TD children, and that head, trunk, and feet represent the maximum classification with an accuracy of 82.98%. Regarding stimuli, visual condition showed the highest accuracy (89.36%), followed by the visual-auditive stimuli (74.47%), and visual-auditive-olfactory stimuli (70.21%). Finally, the head showed the most consistent performance along with the stimuli, from 80.85% in visual to 89.36% in visual-auditive-olfactory condition. The findings showed the feasibility of applying machine learning and virtual reality to identify body movements’ biomarkers that could contribute to improving ASD diagnosis.

## 1. Introduction

### 1.1. Autism Spectrum Disorder and Repetitive Behaviors

Autism spectrum disorder (ASD) is a neurodevelopmental disorder mainly based on impairments in social communication and interactions’ abilities and on the presence of restricted, repetitive patterns of behavior, interests, or activities [1]. It affects 1 in 160 children [2] and its symptomatology tends to appear from two to four years old, although in some cases it is possible to detect in six months old toddlers [3,4]. ASD studies primarily examine the weaknesses in social interaction abilities’, and less on the stereotyped and repetitive motor behaviors that also affect educational, social, and daily life [1,5]. 

Repetitive behaviors (RBs) are defined as heterogeneous observable motor stereotyped or repetitive sequences characterized by rigidity, invariance, inappropriateness, and being purposeless [6,7]. RBs can occur currently with small changes of the routine or in presence of new and unknowns stimuli to reduce subjective arousal and to cope with unfamiliar events, to maintain homeostasis [8,9,10]. 

Furthermore, RBs can be classified into two groups: common behaviors and complex behaviors [11]. Common behaviors are for example nail-biting, thumb sucking, and hair twirling and they tend to be also frequent in typical development population (TD), particularly situations that might cause stress or anxiety [12]. On the other hand, complex behaviors include more complex stereotypies, such as flapping hands, fingers wiggling, head spinning and banging, stamping the feet, and high levels of head movement and body rocking. Although complex RBs are possible to find in TDs, complex head spinning and banging, arm flapping, finger wiggling, and body rocking are mostly related to ASD; indeed, ASD individuals tend to exhibit RBs more frequently and severely than age-matched TD controls [13,14,15]. 

Several studies have demonstrated the high presence of RBs behaviors in ASD, ranging from 60% to 100% of cases [13,14,15]. The relevance of this study lies in understanding if movement features and frequency can be used as a diagnostic biomarker to classify children with and without ASD.

### 1.2. Traditional Assessment in ASD: Advantages and Limitations

Traditional ASD assessment and diagnosis involve qualitative and quantitative measures, such as semi-structured behavioral task observations (Autism Diagnostic Observation Schedule, ADOS) [16] and structured interview (Autism Diagnostic Interview-Revised, ADI-R) [17]. 

ADOS consists of several structured and semi-structured tasks on communication, use of imagination, social interaction, play, and restrictive and repetitive behavior analysis. The examiner introduces to the child one task at a time, observing whether ASD symptoms are manifested [16]. ADI-R is a semi-structured interview for family caregivers, who answer to questions related to communication, social interaction, and restricted, repetitive, and stereotyped behaviors [17] (see Materials and Methods for a detailed description of both ADOS and ADI-R). Since in ADOS examiner has to judge a child’s performance giving scoring and rating, evaluation relies on the examiner’s expertise and subjectivity; likewise, in ADI-R ASD diagnosis is based on caregivers’ reports rather than on objective evaluation. Although these measures have been always considered as the gold standard for ASD assessment [18], they present some limitations [19,20]. 

Regarding ADOS and ADI-R, limitations are related to the absence of both objective assessment methods and the ecological validity of the setting. Examiners need to be trained and prepared to avoid inappropriate task presentation and administration, which might cause symptoms over- or under-interpretation, providing misleading outcomes [21,22]. Moreover, traditional assessment methods might not tap and conceal compensatory capabilities that have been already developed by the child [23], and social desirability [24] might affect responses veracity to tests. Because of social desirability bias, individuals might respond to tasks or questions in a manner conceived as favorable by others [25]. Likewise, in semi-structured interviews, such as ADI-R, family caregivers might report differently certain child’s behaviors according to their interpretation and will [26]. 

Furthermore, traditional ASD assessment takes place in settings that lack ecological validity (i.e., laboratory) [27,28,29], and results do not mirror performance in real life [30,31]. Indeed, in ASD assessment at a laboratory, children might have learned how to behave according to specific rules and scripts [32]. Concerning RB traditional assessment limitations, direct observation consists of watching individual behavioral sequences, and several weaknesses affect measure reliability, such as difficulties in observing high-speed RBs, analyzing two concomitant RBs, detecting RB sequence beginning, ending, and environmental antecedents [18]. Paper-and-pencil rating scales report RBs frequency and intensity from caregivers’ general observations and impressions, yielding objective methodological issues related to self-report measures, that, as well as in ADI-R, are not accurate and do not properly tap individual RBs characteristics [33,34]. Finally, video-based RB assessment is an off-line RB coding procedure made by experts. Although a video-based procedure is more reliable than direct observation methods and paper-and-pencil procedures, it is laborious, and it takes a long time [33]; moreover, the examiner’s coding ability depends on individual levels of training and expertise [21]. 

To overcome the lack of ecological validity, and to avoid the use of subjective observational diagnostic methods in ASD, there is a need to automatically quantify and assess RBs, that could be fulfilled using technology [35,36]. Indeed, item-independent methods can grant accurate estimation of RB incidence and co-occurrence [9,35,36], and new technologies, such as virtual reality, can provide more ecological validity and controlled methodologies.

### 1.3. Implicit Methods: Biomarkers as Supports for ASD Assessment

Recent advances in social cognitive neuroscience (SCN) shed light on how humans analyze and report beliefs, feelings, and behaviors [37]. SCN is a research area that studies biological processes and related cognition-based elements [38], and it is showing that social interactions rely on implicit psychophysiological processes uncontrolled by conscious awareness [39]. 

Implicit measures tend to assess automatic biological processes outside conscious awareness [38] that ensue from the interaction with environmental external stimuli and their internal processing. Such biomarkers are a valid alternative to explicit measures, which cannot tap implied brain processes on their own [40]. Thus, to overcome explicit measure weaknesses in the ASD diagnosis, more recent research has included biomarkers along with traditional assessment techniques [41,42]. 

To date, the available utmost biomarkers to study unconscious processes are the electrodermal activity (EDA) [43,44], the functional magnetic resonance imaging (fMRI) [45], the functional near-infrared spectroscopy (fNIRS) [46], the electroencephalography (EEG) [47], the eye tracking [48], and the heart rate variability (HRT) [49]. In ASD assessment, fMRI research showed that ASD is related to hyperactivity in neural activation and alterations in the cingulate posterior cortex and portions of the insula [50]; whereas EEG research suggested that in social context ASD individuals exhibit greater activity in the left hemisphere [51]. Furthermore, recently developed technological tools, such as cameras and/or sensors, allowed the detection and classification of behavioral biomarkers, as body movements [52].

Initial studies on RBs with such devices showed that more automatic and objective assessment is possible, achieving the recognition of RBs [19,33,36,53,54,55,56,57,58]. For instance, to disentangle RBs from other movements in ASD, three wireless accelerometers placed on six ASD children’s wrists and chest in two different settings were able to accurately identify respectively in lab and classroom 86% and 89.5% of spontaneous hand flapping and body rocking cases [33].

Also, RGB color camera equipped with depth sensor and microphones array for simulated-hand flapping discernment was used in a laboratory setting; the application of the dynamic time warping (DTW) algorithm on-camera recorded data showed that it can recognize and isolate all simulated hand flapping instances, leading to claim that RBs detection is possible even involving sensors that not have necessarily to be worn [19].

### 1.4. Repetitive Behaviors Recognition in ASD

To overcome issues related to traditional RB assessment [18,19], a new research area about movement analysis based on video recordings and automatic tagging has emerged [19,33,36,53,54,55,56,57,58].

The first attempt has been developing systems that involve video recordings and accelerometers placed on the subject’s body to register movements and to transmit data via wireless [35,36,54,59,60,61].

These measures have yielded promising results in RB classification, although some studies have involved typical population and not ASD individuals [61], and to our knowledge, no studies have assessed RBs involving quantitative methods and have compared them between ASD and other clinical populations. However, accelerometer and video-based methods are expensive in terms of time and effort, and ASD children might feel as uncomfortable wearing accelerometers. For this reason, this work involved an RGB-D camera for real-time analysis. RGB-D camera is a video recording device able to augment the conventional image with depth information, related to how much the recorded moving body is far from the sensor.

Owing to deep learning and big data techniques [62,63,64], algorithms have been developed that can estimate real-time subjects’ posture and categorize automatically their movements. Over the last decades, great progress has been made in posture estimation and modern technologies, allowing classifying postures regardless of the worn clothes and the considered point of view [65,66,67,68,69]. Furthermore, machine-learning methods (ML) are improving the predictive value of motor behavioral biomarkers’ measures in ASD, enhancing the development of objective measures in the diagnostic standpoint [70,71]. For example, Crippa et al. (2015) developed a ML to discriminate preschool children with ASD from children with typical development using a simple upper-limb reach-to-drop task. The resulting model showed an accuracy rate of 96.7%, suggesting that ML can be a useful method of classification and discrimination in the diagnosis process [70].

### 1.5. Use of Virtual Reality in ASD

Virtual reality (VR) is a three-dimensional computer-generated environment that allows users to experience simulated and unreal worlds. VR provides ecological validity to experienced situations and consequent users’ reactions, becoming promising in psychological assessment, training, and treatment [72,73].

Over the last two decades, the VR market has deeply grown because of the broad amount of enterprise birth and consequent wide offer of virtual devices. Overall, head-mounted displays (HMDs) are the most important, affordable, and available in the VR market [74]. However, a different VR device has been suggested as more suitable to our target (ASD children): the CAVE-Automatic Virtual Environment (CAVE™) that is a semi-immersive room where 3 to 6 rear-projected surfaces are installed [74,75,76,77,78].

As a semi-immersive system, CAVE™ overcomes users’ risk to experience cyber-sickness that is the possible user’s discomfort because of sensory-motor incongruence and cognitive dissonance in the virtual world [79]. Furthermore, specifically to ASD children, CAVE system can overcome the significant restrictions of HMDs that, on one hand, are not suitable for small heads, and on the other, can affect and worsen their sensory and cognitive difficulties [80,81]. Previous studies on feasibility, safety of use, and learning skills of CAVE environments in children with ASD have showed no significant negative effect differences between ASD and TD children and improvements in various skills (e.g., pedestrian crossing) [82].

Regardless of the involved technology and the brand, VR systems share three main features: immersion, interaction, and sense of being present in the environment [83,84,85,86,87,88]. Immersion refers to system capability to isolate the user from reality [86,88]. Interaction allows users to interact with virtual objects in real-time through control sticks or gloves, providing engagement, motivation, and fun [84,89]. Sense of presence is a consequence of immersion and real-time interaction, and it is defined as the psychological feeling related to the sensation of being physically in the virtual environment, even though the awareness of not being there [88,90,91,92,93,94]. Finally, another feature to consider, less addressed in studies and especially in ASD population, concerns the perception and interaction with virtual agents. Virtual agents have been mainly used in social trainings and interventions showing positive results on skills in ASD children [82]. These positive results suggest that virtual agents are perceived not as cartoons or passive objects to watch, but actively as an intentional being that wants to communicate with the child and with mutually directed behaviors. To our knowledge, one study examined the perception of ASD children in the interaction with a virtual agent for performing a task (pick up flowers) [95]. The quantitative data (accuracy and reaction times) showed that ASD children could complete the task and the qualitative data showed that most of the ASD children perceived the virtual agent as an intentional being with mutually directed behavioral intentions, able to engage and motivate the ASD peer.

There is a great involvement of VR in psychological ASD treatment and it taps all important field macro-areas, such as clinical psychology, neuropsychology, and cognitive and motor rehabilitation [75,85,96,97,98,99,100,101,102,103,104,105]. Specifically, treatment studies, applying VR to ASD, mainly referred to social competence [101,102,104], emotional recognition [103], and anxiety and phobias [105] showing initial positive effects of this technique. Regarding ASD VR assessment, it has been less addressed and mainly focused on social communication and interaction symptoms. For example, it has been observed that during a virtual interview about personal life, ASD children looked less to social avatars than their TD control peers, identifying correctly 76% of ASD cases [102]. Also, ASD children made atypical social judgments on the kindness of faces photographs in a virtual environment compared to TD controls [103]. Despite VR potential in ASD assessment has already been strengthened [75], to our knowledge, no one has investigated whether it is possible to disentangle ASD in a VR experience using RBs movement analysis.

Starting from these premises, the main aim of this study is to discriminate ASD children from children with typically developing through body movements’ data analysis in a multimodal VR experience, characterized by three stimuli: visual, auditory, and olfactory. Applying ML methods to the dataset, we explored: (a) If through movement data it is possible to discriminate between the two populations; (b) which body parameters better discriminate between the two populations; and (c) which virtual stimuli condition better discriminate body parameters between the two populations. 

## 2. Materials and Methods

### 2.1. Participants

This study included a sample of 49 children between the ages of 4 and 7 years; 24 children with a diagnosis of ASD (age: 5.13 ± 1.35; male = 21, female = 3) and 25 with a typical development (TD) (age: 4.86 ± 0.91; male = 16, female = 9).

The ASD group sample was provided by the Development Neurocognitive Centre, Red Cenit, Valencia, Spain. TD and ASD participants presented an individual assessment report that included the ADOS-2 and ADI-R tests [16,17,21]. TD group was recruited by a management company through mailings to families. 

To participate in the study, family caregivers received written information about the study and were required to give their written consent. Ethical Committee of the Polytechnic University of Valencia approved the study. The study procedure was in accordance with the ethical standards of the institutional and national research committee and with the 1964 Helsinki declaration and its later amendments or comparable ethical standards. 

### 2.2. Psychological Assessment

The following test and scales have been administered to participants and their family caregivers.

#### 2.2.1. Autism Diagnostic Observation Schedule (ADOS-2)

The Autism Diagnostic Observation Schedule (ADOS) [16,21] is a semi-structured set of observation tasks that measure autism symptoms in social relatedness, communication, play, and repetitive behaviors. A standardized severity score within these domains can be calculated to compare autism symptoms across the different modules, which differ in age and linguistic level. From the trained psychologist observation of these behaviors, the items are scored between 0 to 3 (from no evidence of abnormality related to autism to definitive evidence) and from the sum of scores are obtained two specific indexes (social affectation and restricted and repetitive behavior) and an ASD’s global total index. The ADOS-2 presents excellent test-retest reliability (0.87 for the social affectation index, 0.64 for the repetitive behavior index, and 0.88 for the total global index). In the study, the assessment was performed using module 1.

#### 2.2.2. Autism Diagnostic Interview-Revised (ADI-R)

The ADI-R [17] is a semi-structured interview for family caregivers, designed to provide the developmental history framework for a lifetime to detect the presence of ASD for individuals from early childhood to adult life. It consists of 111 questions with three separate domains: communication, social interaction and restricted, repetitive and stereotyped behaviors. The answers are scored on a 0–3 Likert scale, from the absence of the behavior to a clear presentation of the determined behavior. ADI-R presents a high test-retest reliability ranging from 0.93 to 0.97.

### 2.3. The Multimodal Virtual Environment (VE) and the Imitation Tasks

The multimodal VE consisted of a simulated city street intersection and was divided into three experimental stimuli conditions: visual (V), visual-auditory (VA), and visual-auditory-olfactory (VAO) (Figure 1a). 

First, in the V condition, a boy’s avatar appeared from the left side of the surface CAVE™, walking to the middle of the virtual environment, where he stopped and waved to say hello to the participant three times, just before to leave the virtual scene disappearing from behind (Figure 1b).

Next, a girl’s avatar appeared in the center of the surface CAVE™, walking to the right of the virtual scene, where she also stopped and repeated the three waves to say hello to the participant, just before leaving the virtual scene disappearing from behind. This sequence was identically repeated three times. 

Consecutively, in the second VA stimuli condition, the same avatars appeared in the same first order from the same directions, and they danced over an animated disco song for 10 s three times. 

Finally, in the last condition (VAO), the same avatars from the two previous conditions, in the same order and from the same directions, bit a buttered muffin, accompanied with an artificial butter odor (Figure 1c). In the three stimuli conditions, participants were asked to imitate the actions performed by the avatars. Specifically, in the first virtual condition, they should wave three times, dance three times in the second virtual condition, and they should imitate the action of biting a muffing three times. 

The selection of the stimuli and the gradual exposition to the three stimuli conditions depended on the hyper-and-hypo sensitivities to sensory stimuli of the ASD population. More in detail, with respect to visual and auditive stimuli (e.g., bright lights or noisy sounds), ASD population presented a hypersensitivity [106,107]; conversely, they present hypo-sensitiveness to olfactive stimuli [108,109]. Sensory hyper-and-hypo sensitivities consequently can affect the information processing in ASD and it has been suggested it may cause RBs [110,111]. 

The Institute for Research and Innovation in Bioengineering (i3B) of the Polytechnic University of Valencia (UPV) developed the 3D modeling. The environment was projected inside a three surfaces Cave Assisted Virtual Environment (CAVE™) with dimensions of 4 × 4 × 3 mt. It was equipped with three ceiling ultra-short lens projectors, which can project a 100° image from just 55 cm and a Logitech Speaker Z906 500W 5.1 THX sound system (Logitech, Canton of Vaud, Switzerland) (Figure 2).

### 2.4. The Olfactory System

Olorama Technology™ (www.olorama.com) a wireless freshener delivered the olfactory stimuli. It can encompass until 12 scents arranged in 12 pre-charged channels, which can be selected and triggered by employing a UDP packet. The device includes a programmable fan time system that controls the duration and intensity of the scent delivery. In the VAO condition, we used a butter scent to evocate the real muffin smell. The scent valve was opened all the time during the last stimuli condition (VAO). 

### 2.5. Experimental Procedure

First, the family caregivers of participants were informed about the general objectives of the research, and, before the experimental session, the setting was also shown and explained to them.

Second, the participant was accompanied in the CAVE™, by the researcher, and by his or her family caregiver according to the child’s needs. The participant was placed in the middle of the virtual room, standing in front of the central surface at 1.5 m. The presentation order of VR stimuli conditions for all participants was: visual, visual-auditory, and visual-auditory-olfactory. Before each stimuli VR condition, a two-minute baseline was recorded, and subsequently, the stimuli VR experience condition was presented. The presentation order was maintained the same for all participants to avoid and prevent sensory overload and stress that they could experience.

The total duration of the experiment was of 14 min, and each stimulus condition lasted 2 min and 40 s. Movement recording was turned on during the virtual experience. The researcher monitored the child state during the entire experiment.

### 2.6. Behavioural Motor Assessment and Data Processing

In the experiment design, it was proposed to use an efficient method to estimate the pose in real-time using an RGB-D camera. The participant’s experiment was recorded using an Intel^®^ RealSense™ camera D435 (FRAMOS, Munich, Germany) and Intel RealSense SDK 2.0 (Intel RealSense Technology, Santa Clara, CA, USA) with the cross-platform support. This camera has a depth sensor that uses stereo vision to calculate it. It works like a USB and is equipped with a pair of depth sensors, an RGB sensor, an infrared project a great global image obturator (91.2° × 65.5° × 100.6°).

The detection of the body joints in each frame of the recording was calculated using the deep learning algorithm OpenPose [112], which includes the 2D position related to the video and a confidence level for each joint identification between 0 and 1. The skeleton (Figure 3) includes 25 joints that can be divided into different parts of the body: head (0 nose, 15/16 eyes and 17/18 ears), trunk (1 neck and 8 mid hip), arms (2/5 shoulders, 3/6 elbows and 4/7 wrists), legs (9/12 hips, 10/13 knees, 11/14 ankles), and feet (19/22 big toes, 20/23 small toes and 21/24 heels). After detecting the skeleton, the 3D position of each joint during the experiment was obtained using the depth information of the camera. A computer with an Nvidia GTX1060 graphics card with the NVIDIA Pascal architecture capable of executing neural networks very efficiently in a compact size was used. 

After extracting the 3D position of the body’s joints from all the experiments, the data recording was segmented for each stimulus condition, excluding the samples of the joints that have a confidence below 0.5. The displacement of the joints was computed using the Euclidean distance between consecutive frames. Finally, the level of movement of a joint during a stimulus was characterized by computing the mean of all displacement.

### 2.7. Statistical Analysis

First, to characterize behavior differences between ASD and TD children on each stimuli condition, we analyzed the movement frequency of each joint. Since data followed a non-normal distribution (Shapiro-Wilk test: *p* < 0.05), Wilcoxon signed-rank tests were applied. 

Second, we applied a set of machine learning models to analyze if the frequency of the movement could discriminate between ASD and TD children. The body was divided into five parts: head (joints 15, 16, 17, and 18), trunk (joints 1 and 8), arms (joints 2, 3, 4, 5, 6, and 7), legs (9, 10, 11, 12, 13, and 14), and feet (19, 20, 21, 22, 23, and 24). The machine learning analysis based on 24 features-based dataset included the five body parts, one related to the whole body, the three stimuli conditions (V, VA, VAO) and one related to the entire experiment. Because of a large number of features, a reduction strategy was adopted to decrease the dimensions in each dataset. Principal component analysis method (PCA) was applied to select features that explain 95% of the variability of the dataset. Finally, a supervised machine-learning model was developed using the PCA features in each dataset.

To implement the models, support vector machine (SVM)-based pattern recognition with a leave-one-subject-out (LOSO) cross-validation procedure has been applied [113]. Within the LOSO scheme, the training set was normalized by subtracting the median value and dividing by the median absolute deviation over each dimension. In each iteration, the validation set consisted of one specific subject and it was normalized using the median and deviation of the training set. We used a C-SVM optimized using a Gaussian Kernel function, changing the parameters of cost and gamma using a vector with seven parameters logarithmically spaced between 0.1 and 1000. Also, a SVM recursive feature elimination (SVM-RFE) procedure was included in a wrapper approach. RFE was performed on the training set of each fold and we computed the median rank for each feature among all folds. In particular, a nonlinear SVM-RFE was implemented, which includes a correlation bias reduction strategy in the feature elimination procedure [114]. To analyze the performance of the models, a set of metrics were considered: accuracy, i.e., percentage of subjects who correctly recognized, true positive rate, i.e., percentage of actual ASD subjects recognized as ASD, true negative rate, i.e., percentage of actual control subjects recognized as controls, and Cohen’s kappa, which describes the performance of the model from 0 to 1, 0 being a random class assignation and 1 a perfect classification. The model was optimized to achieve best Cohen’s kappa. The algorithms were implemented using Matlab© R2016a and LIBSVM toolbox [115].

## 3. Results

### 3.1. Analysis of Total Movement

Figure 4 shows the body joints’ significant differences obtained applying Wilcoxon signed-rank. In V stimuli condition, 4 joints of the legs, 2 of the head and 1 of the trunk presented higher movements in ASD population than TD children. In VA stimuli condition, 1 joint of the legs and 2 of the head also presented higher movements in ASD population than TD children. Also in VAO stimuli condition, 3 joints of the head presented the same tendency. Conversely, 1 joint of the head (16) presented lower movement in ASD population in the visual condition baseline (BL_V) and in visual-auditive-olfactive condition baseline (BL_VAO).

### 3.2. ASD Classification Performance

Table 1 and Table 2 show the performance of the computed machine learning models, considering the combination of a set of joints and virtual stimuli conditions. Regarding body parameters, the models including the head and trunk joints presented an accuracy of 82.98% and unbalanced confusion matrices. Conversely, the model that uses the feet joints present the same accuracy (82.98%) and a balanced confusion matrix. In addition, the models including arms and legs movements achieved a lower accuracy than the other body joints (74.47% and 72.4% respectively). Finally, the model including all joints and virtual stimuli conditions showed the lowest accuracy (70.21%) and the lower true positives (45.45%). 

Regarding the influence of the virtual stimuli conditions, the V presented the highest accuracy in the study (89.36%), showing that it is the most relevant stimuli of the experiment. In the VA stimuli, the accuracy decrease to 74.47%, and in VAO to 70.21%. Furthermore, the head joints showed the most consistent performance along with the stimuli, from 80.85% in V to 89.36% in VAO, suggesting that it is the most important part of the body to discriminate ASD population. The trunk joints present also a consistent accuracy, from 70.21% in V to 76.60% in VAO, but the discriminate performance is considerably lower than using the head. All the models used three features or less after applying the automatic feature selection procedure.

## 4. Discussion

ASD is diagnosed according to qualitative clinician judgments, based on symptoms, through semi-structured observations and interviews (ADOS; ADI-R). Given the qualitative nature of the traditional tools, researchers have focused on improving methods of diagnosis and assessment, pointing out the predictive value of behavioral biomarkers, as more objective and quantitative measures. This study aimed, first, to compare tracked body movements data during a multimodal VR stimulation between ASD and TD children. Second, it was verified which body areas might be relevant for the discrimination between the two populations. Third, the study investigated which virtual stimuli condition better discriminate body areas between the two populations. To reach these aims, we applied a machine learning procedure [70] to body movements’ analysis of a multimodal VR experience, composed of three stimuli conditions: visual, visual-auditive, and visual-auditive-olfactory. 

The study included a preliminary analysis of the frequency distribution of body movements to investigate the differences between groups and a broad set of supervised ML models combining body parameters and stimuli conditions to evaluate the discriminability between ASD and TD children using movement. Results can be discussed on four points: (1) The significant differences between the two groups on body movements; (2) machine learning methods on body movements and features used; (3) the influence of stimuli conditions; and (4) limitations and future studies.

### 4.1. Body Movement Parameters’ Differences between Groups

The first aim was to identify differences in terms of body movements between children with ASD and TD children. Figure 4 showed significant differences in15 body joints and 13 of them have showed that ASD children present more body movements than TD children. In particular, 4 legs’ joints, 2 head’s joints and 1 trunk’s joint in V condition, 1 legs’ joints and 2 head’s joints in VA condition, and 4 head’s joints in VAO condition showed higher movements, suggesting that ASD children have performed more head and legs’ movements than TD children during the imitation tasks. Previous studies that showed that head tilting, legs flapping, as well as bilateral repetitive movements involving legs walking represent two or more movements features related to ASD children confirm the presence of larger body movements’ in ASD than in TD children [8,10]. However, only one joint in the head showed higher movement in TD children than ASD children during the baselines of VA and VAO. This result is partially in opposition to the scientific literature but as mentioned in the introduction, common and complex body movements are also present in TD children. To improve the quantitative methods to discriminate ASD from TD children using body movements’ frequencies, we applied machine-learning techniques developing 24 models, combining body joint parts and virtual stimuli conditions. In the next section, we deeply discuss the results.

### 4.2. Machine Learning Methods on Body Movements and Features Used

Concerning the recognition using different parts of the body, the best classification accuracy reached 82.98% for the head parameter by using only one feature selection method, and the same accuracy was achieved for body and foot parameters by using two features’ selection method, independently by the specific stimuli condition. Interesting results have also resulted from the model including arms and legs that achieved a classification accuracy of 74.47% and 72.34%, respectively. The results are consistent with the scientific literature that has identified head spinning and banging, body rocking, and stamping the feet, three of the main stereotypies and repetitive movements are related to ASD [8]. Furthermore, these results showed that ML could provide an effective solution to best describe body movements’ differences of participant groups’ classification during imitation tasks, while the traditional statistical methods (e.g., mean age comparisons) could fail to deal with such complex tasks. To our knowledge, only two previous studies have investigated the predictive value to discriminate between children with and without ASD using an imitation task [70,71]. Both studies focused on the analysis of arm and hand movements during a simple reach-to-drop imitation task, reaching a maximum classification accuracy of 96.7% and 93.8%, respectively. Our study focused on a more complex real-simulated imitation task, composed of three imitation subtasks, waving, dancing, and eating and tracking all body movements.

Thus, the present findings show the feasibility and applicability of a ML method for correctly classifying preschool children with ASD based on real-simulated imitation tasks. In the standard diagnosis of toddlers it is particularly difficult to evaluate repetitive and stereotyped behaviors. The application of technologies, such as cameras and sensors, might have potential clinical applications to support diagnosis providing accurate quantitative methods along with the qualitative traditional methods. Finally, the use of cameras and sensors are more convenient, less expensive, and less invasive technologies than fMRI and EEG to implement in clinical settings. 

### 4.3. The Influence of Stimuli Conditions

Regarding the virtual stimuli conditions, head showed to be the core body movement in the groups’ classification during the three stimuli conditions, achieving the best classifications’ accuracy of 89.36% in VAO condition, 82.98% in VA condition, and 80.85% in V condition. These results suggest that the greater the sensory stimulation, the greater the stereotypies and repetitive movements, allowing to discriminate children with ASD from TD children. The different classification accuracy among the three stimuli conditions is consistent with previous studies on the sensory sensitivity overload of ASD patients to multimodality stimuli [116,117,118,119]. Specifically, ASD patients have shown lower-functioning abilities to filter process and integrate simultaneous information compared to TD subjects [116]. For example, during multiple auditory tones, tasks matched with single visual-flash stimulus ASD patients perceived more flashes than those are presented [117]. Furthermore, EEG studies on ASD patients showed that the amplitude is higher and the latency in the response is delayed than TD subjects [118,119]. Other multimodal sensory stimulations achieved a good accuracy classification between the two groups, resulting in stereotypes and repetitive movements of the trunk (VAO: 76.60%), legs (VAO: 74.47%), arms and feet (VA: 78.72%). Finally, arms and feet movements showed a high accuracy of 87.23% and 78.72 respectively in the classification between groups during the unimodal visual condition.

In general, V condition including all the body achieved the best accuracy (89.36%), including a 100% true positive rate and using only 1 feature of PCA, in contrast to VA (74.47%) and VAO (70.21%). These results, on one hand, highlight that real-simulated imitation activities that can occur in daily life, such as waving (V), dancing (VA), and eating (VAO) allow evaluating body movements discriminating ASD and TD children. Previous studies on discriminating ASD and TD children using imitation tasks have used simple reach-to-grasp and drop arm movement tasks [70] or a hand movement task to target in laboratory settings [71] and our study aimed to investigate whole-body movements in more complex real-simulated scenarios and tasks. 

On the other hand, the predominance of the visual influence can depend on the fact that V stimuli was the first condition and could have generated a wow-effect in the subjects. Finally, the model that includes all the data achieved an accuracy of 70.21% with a 45.45% of true positive rate, showing that the feature selection procedures have a critical role to achieve good performance in biomarkers development. This result could suggest that the implemented virtual tasks could fail in detecting RB behaviors. 

### 4.4. Limitations and Future Studies

Despite the promising results, some methodological limitations of the present exploratory study should be reported. The main limitation is the limited sample sizes of participants for each group. Future studies on larger sample sizes could allow validating the ML method, providing also the possibility to test the model. As ASD is a heterogeneous condition, the possibility of training the model with larger groups of ASD and children with typical neurodevelopment would be useful to improve the generalization of the model to a wide condition range of ASD. Also, other comparisons studies, including other neurodevelopment disorders with movement impairments, as the attention deficit hyperactivity disorder, could be valuable by ML for understanding if these groups present similar body movements that are distinct for the different neurodevelopment conditions (e.g., intellectual disability [120]. Indeed, taking also into account different neurodevelopment conditions could improve the specificity of the classifiers for ASD rather than for neurodevelopment disorders in general. 

Furthermore, the experimental groups were not matched on measures related to the intelligence quotient performance, cognitive abilities, and motor comorbidities (e.g., dyspraxia), limiting the confidence that the experimental differences observed in the experiment were due to a diagnosis of ASD, and not to differences in other cognitive or non-cognitive factors. Future studies will consider matching these factors in order to improve the reliability and validity of the model results. 

Finally, regarding the stimuli conditions, the order of presentation (V-VA-VAO) was not counterbalanced because of the hyper-sensitiveness of ASD children to multiple stimuli [105,106,107,108]. Future studies including counterbalanced conditions following high and low children functioning are needed.

## 5. Conclusions

The present study represents, to our knowledge, the first attempt to discriminate ASD children from children with typical neurodevelopment using body movements parameters of an imitation VR task and machine learning. The significant predictive values of our classification approach might be valuable to support ASD diagnosis, as well as the use of more objective methods and ecological tasks VR-aided along with traditional assessment. 

## Figures and Tables

**Figure 1 jcm-09-01260-f001:**
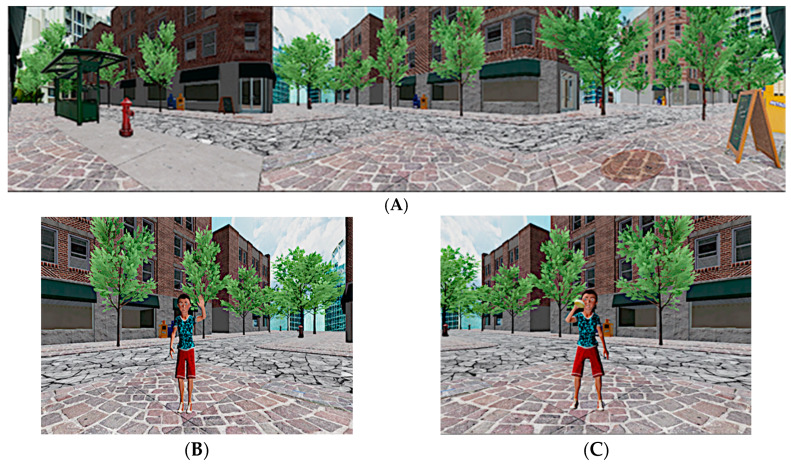
The virtual environment. (**A**) City street intersection; (**B**) visual (V) condition, boy’s avatar saying hello; (**C**) visual-auditory-olfactory (VAO) condition, boy’s avatar eating a muffin.

**Figure 2 jcm-09-01260-f002:**
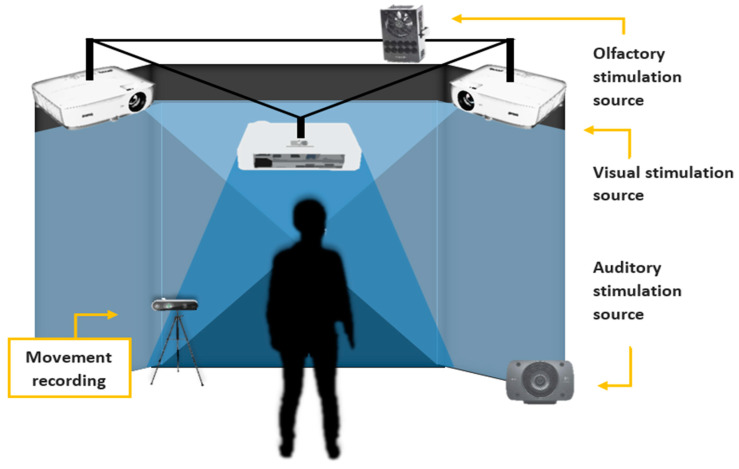
Experimental setting.

**Figure 3 jcm-09-01260-f003:**
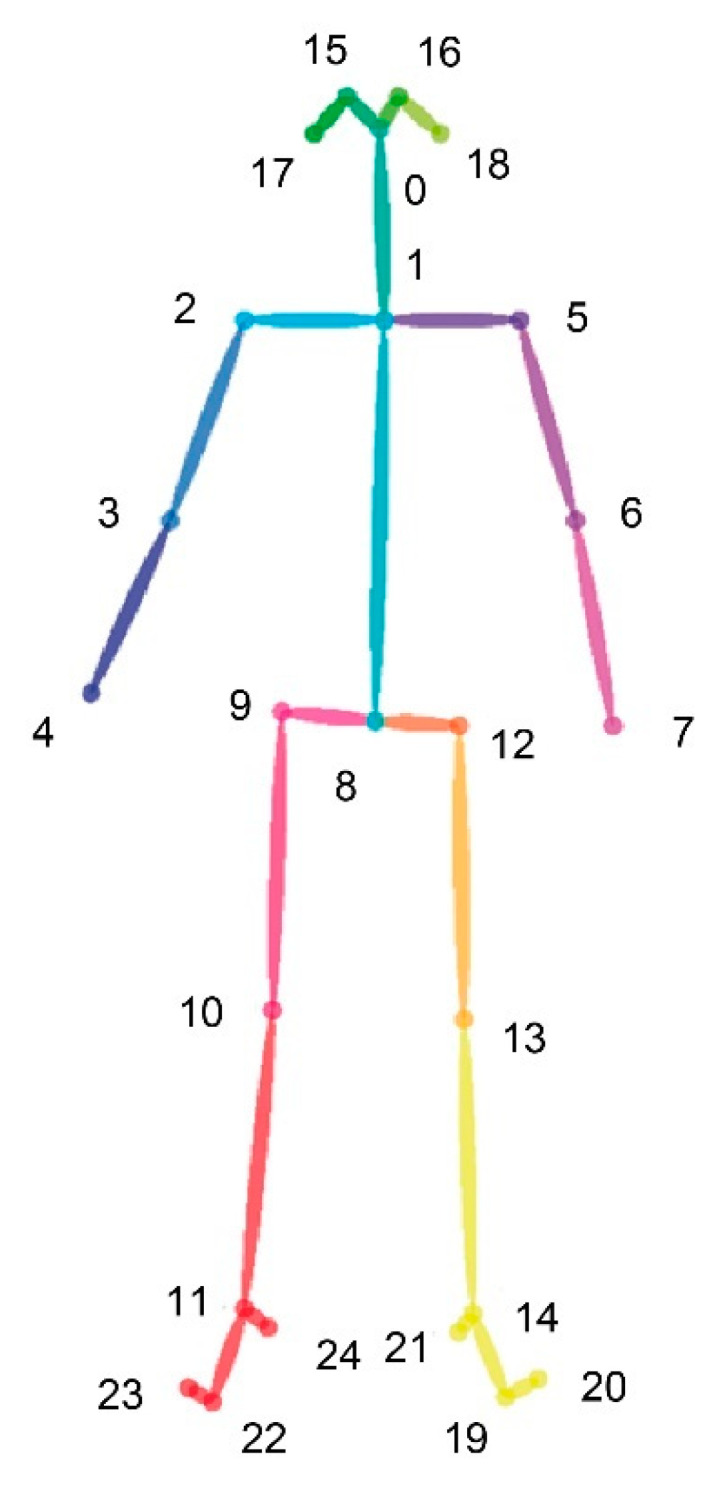
Joint virtual disposition.

**Figure 4 jcm-09-01260-f004:**
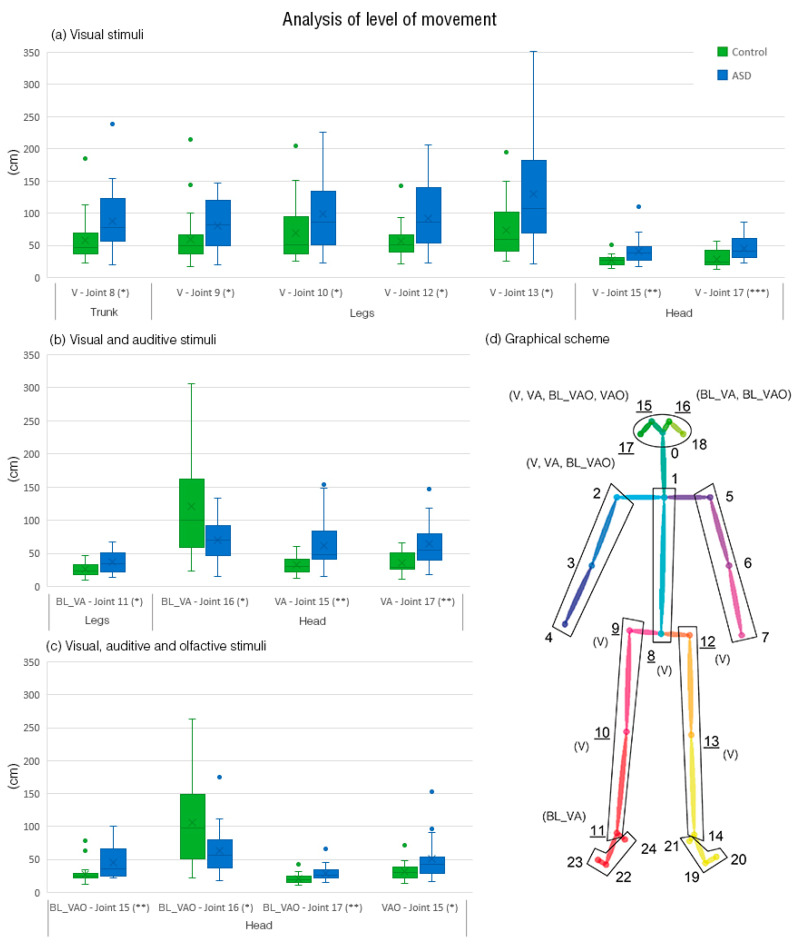
Analysis of the level of movement of joints that presents statistical differences: (**a**) Boxplot of joints in visual stimuli; (**b**) boxplot of joints in visual and auditory stimuli; (**c**) boxplot of joints in visual, auditory, and olfactory stimuli; (**d**) scheme of the joints including the stimuli where differences are found. Note. * *p* < 0.05. ** *p* < 0.01. *** *p* < 0.001.

**Table 1 jcm-09-01260-t001:** Overview of the accuracy of each model considering the stimuli and set of joints.

		Stimuli			
		V	VA	VAO	All
Set of Joints	Head	80.85%	82.98%	89.36%	82.98%
	Trunk	70.21%	72.34%	76.60%	82.98%
	Arms	87.23%	78.72%	65.96%	74.47%
	Legs	61.70%	63.83%	74.47%	72.34%
	Feet	78.72%	78.72%	65.96%	82.98%
	All	89.36%	74.47%	70.21%	70.21%

**Table 2 jcm-09-01260-t002:** Detailed level of autism spectrum disorder (ASD) recognition of each model including accuracy, true positive rate (TPR), true negative rate (TNR), Cohen’s Kappa and PCA featured selected.

Stimuli	Set of Joints	Accuracy	TPR	TNR	Kappa	PCA Features Selected
All	All	70.21%	45.45%	92.00%	0.39	1/20
V	All	89.36%	100.00%	80.00%	0.79	1/14
VA	All	74.47%	59.09%	88.00%	0.48	2/12
VAO	All	70.21%	63.64%	76.00%	0.40	3/12
All	Head	82.98%	100.00%	68.00%	0.67	1/11
All	Trunk	82.98%	63.64%	100.00%	0.65	2/8
All	Arms	74.47%	90.91%	60.00%	0.50	1/15
All	Legs	72.34%	68.18%	76.00%	0.44	3/8
All	Feet	82.98%	81.82%	84.00%	0.66	2/13
V	Head	80.85%	68.18%	92.00%	0.61	1/6
V	Trunk	70.21%	81.82%	60.00%	0.41	1/3
V	Arms	87.23%	72.73%	100.00%	0.74	1/8
V	Legs	61.70%	45.45%	76.00%	0.22	1/5
V	Feet	78.72%	54.55%	100.00%	0.56	2/6
VA	Head	82.98%	63.64%	100.00%	0.65	3/6
VA	Trunk	72.34%	72.73%	72.00%	0.45	1/3
VA	Arms	78.72%	95.45%	64.00%	0.58	1/8
VA	Legs	63.83%	45.45%	80.00%	0.26	1/4
VA	Feet	78.72%	72.73%	84.00%	0.57	2/6
VAO	Head	89.36%	77.27%	100.00%	0.78	2/6
VAO	Trunk	76.60%	68.18%	84.00%	0.53	2/3
VAO	Arms	65.96%	50.00%	80.00%	0.30	2/7
VAO	Legs	74.47%	68.18%	80.00%	0.48	1/5
VAO	Feet	65.96%	45.45%	84.00%	0.30	2/7

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
