# Peer review of "Machine Learning and Virtual Reality on Body Movements’ Behaviors to Classify Children with Autism Spectrum Disorder"

_jcm, 2020, doi:10.3390/jcm9051260_

Round 1

Reviewer 1 Report

The study described in this manuscript investigates a novel VR procedure to use body movements as a diagnostic predictor of ASD. The authors correctly point out that truly objective measures, particularly those that capture biological processes, are not widely used to facilitate the diagnostic process in ASD, which is instead based on descriptions of behaviour often by an untrained observer (i.e. the primary caregiver). The method described in this study, therefore, is both novel and potentially impactful in the clinical setting.

The manuscript itself is very well written and easy to follow, and the study design itself has been carefully thought out. The research involves a simple VR paradigm, with visual, auditory and olfactory stimuli presented during which the participant is asked to imitate what they see. The level of movement of a joint during a stimulus was characterized by computing the mean of all displacement, and differences between ASD and TD groups are presented and the predictive accuracy of the movements to differentiate between ASD and TD groups.

I think this is an important study, and I have no major concerns.

The relatively small sample size is one drawback, but I imagine it was a difficult study to set up.

It would be nice to have some more information about the participants, in terms of cognitive level, comorbidities (e.g. dyspraxia as a relevant co-morbidity comes to mind) and any medication they may have been taking.

How do you know the TD group were TD?

I noticed a couple of grammatical errors. For example, p.7 “Bitted a buttered muffin” > “bit a buttered muffin”, and p.7 “bit a muffing” > “biting a muffin”.

Author Response

Dear reviewer,

Thank you for your revision and positive comments on the manuscript.

Regarding the sample size, we found difficulties to encounter the experimental groups and, as you said, it was a difficult study to set up. Our future objective is to increment the sample size, as well as to improve the methodology, matching on more information about participants (e.g. IQ performance, motor and cognitive control) as you suggest. In the revised manuscript, this consideration has been included in the limitations section of the Discussion. Regarding TD group, line 222 reports the information required.

Reviewer 2 Report

Corrections

Abbreviations:

ln = line in document.

Fig = Figure.

  • Ln 17: remove “allowing”.
  • Ln 19: change “are allowing detection” to “can accurately detect”.
  • Ln 26: change “rely” to “examine”.
  • Ln 48-52: sentence too long; break up with a period.
  • Ln 53: Replace “Besides” with “Furthermore”. Also, change “Besides” in rest of the document. It is too conversational.
  • Ln 55-56: “particularly in situations of distress and anxiety traits [12]; on…” change to “particularly situations that might cause stress or anxiety [12]. On…”.
  • Ln 70: change “consist” to “consists”.
  • Ln 72: change “kid” to “child”, and do so for the rest of the document.
  • Ln 75: swap “exhaustive” with “detailed”.
  • Ln 76: “examiner” to “examines”.
  • Ln 79-81: overly verbose, be more concise.
  • Ln 92-95: “Regarding…setting [27, 29]” overly verbose, be more concise.
  • Ln 103: change “basing on” to “from”.
  • Ln 132: change “allowed detecting and classifying” to “allow the detection and classification of”.
  • Ln 135: what is meant by “majority?” Please rephrase.
  • Ln 154: add a full-stop after “accelerometers”.
  • Ln 158: change “the algorithm makes possible to” to “algorithms have been developed that can”.
  • Ln 185: “glows” to “gloves”.
  • Ln 242: remove table 1, it does not add anything to the paper.
  • Ln 306: is it “RGBD” or “RGB-D”? Keep consistent.
  • Ln 317: The skeleton unit numbers in Figure 3 do not match the units on the skeleton in Figure 4.
  • Ln 332: How do 15,16,17,18 map onto a human head? Please describe in more detail.
  • Ln 333-335: “The combination…features-based.” Please rephrase.
  • Ln 359: What are “BL_VAO” and “BL_VAO”? Please provide more details. Also, these appear in around the head of the skeleton in Figure 4 but in a different order (e.g., “lb_vao”). Please keep these terms consistent and use uppercase.
  • Ln 365: change “(*, **, *** indicate significant differences with p < 0.05, 0.01, 0.001 respectively)” to “ codes: * p < 0.05. ** p < 0.01. *** p < 0.001.”.
  • Ln 383: Table 2 “Set of Joints” looks cramped and untidy.
  • Ln 385: Explain TPR and TNR in the method section alongside the dependent variables.
  • Ln 406: “ `differences ” to “differences”.
  • Ln 461: 78.72%.

Highlights

The paper uses a novel VR setup and machine learning to examine differences in the motor functioning of children with and without an autism-spectrum disorder (ASD). The papers main strengths are in it’s novel method and data analysis techniques.

It is interesting that the detection algorithm is less able to classify children from richer stimuli (e.g., by the inclusion of auditory and olfactory stimulus) presentations. This finding aligns with other sensory data and theories related to sensory integration differences in ASD.  

The paper is also well researched, containing many recent (post 2000) citations from the field of ASD and motor control research.

Suggested amendments

My comments mainly relate to the method. Please see each of these below.

Methodological considerations

Having read the method section I was not entirely certain what instructions were given to parents/guardians and the children participating. Please give thorough details of these instructions. Also, it would be useful what the practice trial involved, and if all the children were motivated to continue after completing the practice trial; children lacking motivation may decide to move their bodies out of boredom rather than in response to the task.  

Justification for use of the CAVE VR system was that it was suggested as a suitable option. Researchers in the field will want more details to why this is the case. Who suggested this – are they a practitioner working with autistic children? Is there something specific about CAVE that makes it useful for ASD research relative to other VR systems? Has anyone used the system before with special or vulnerable populations? What did they find? Please provide more information about the choice of system used.

You have included visual, auditory, and olfactory modalities stimuli. What is the justification for this? With particular reference to the olfactory modality; is there research to show there are differences between TD and ASD children in their sense of smell? How might this affect their RBs? A better explanation for the selection and inclusion the three sensory modalities is required.

The task itself involves imitation of a virtual agent, however there is no literature discussing how virtual agents have been used for this purpose previously. This is important because differences observed in the task might be due to ASD children’s difficulties with theory of mind (e.g., placing themselves in the “minds eye” of the agent) rather than motor coordination per se. That is, there may be artefacts in the data that could be explained by difficulties with theory of mind rather than motor praxis.

With respect to your experimental groups, did you match the groups on any measures (e.g., performance IQ, or motor control) prior to assessment? Matching groups is standard practice in developmental psychology, giving the researchers more confidence that the experimental differences observed in an experiment are due to a diagnosis of ASD, and not differences in other cognitive or non-cognitive factors. If this is not the case, I would include this consideration in the limitations section of the Discussion.   

Summary

This is a well executed study on account of the method used, the data collected and the findings. However, there is a lack of detail in places that make it hard to judge the experimental rigour of the work – especially in the method. Please consider each of the points raised for the present and future work.

Author Response

Dear reviewer,

Thank you very much for your revisions and positive comments and suggestions. Regarding your general required modifications, in the revised manuscript all the changes are tracked, including tables and figures, as you suggested.

With respect to the methodological considerations, thank you for your comments and suggestions. In the revised manuscript, we considered one by one as follows:

  1. Line 305-306: we clarify the instructions given to caregivers before the experimental session.
  2. We use the CAVE system for two reasons:
  3. The age of the participants (from 4 to 7 years old) and the most feasible immersive technological system for these young children is actually only the CAVE because the HMD systems are not suitable for small heads.
  4. HMD systems show significant restrictions especially for children who may already experience sensory difficulties.

Guazzaroni, G. (Ed.). (2018). Virtual and Augmented Reality in Mental Health Treatment. IGI Global.

Wallace, S., Parsons, S., Westbury, A., White, K., White, K., & Bailey, A. (2010). Sense of presence and atypical social judgments in immersive virtual environments: Responses of adolescents with Autism Spectrum Disorders. Autism, 14(3), 199–213. https://doi.org/10.1177/1362361310363283

In the revised manuscript, we extended the justification for use of the CAVE VR system - Line 174-179.

  1. Regarding the visual, auditory, and olfactory modalities stimuli, in section 2.3 - lines: 252 – 258 – we explained more and clarified the use of these stimuli, the differences between TD and ASD children, and the impact that they have on RBs.
  2. Regarding the imitation of a virtual agent, the literature has less addressed this topic and several studies have trained or treated social skills in ASD children using virtual agents showing positive results. We found only one study that investigated this issue more in detail. In the revised manuscript, we reported this issue and the study – Lines 186-196.

Alcorn, A., Pain, H., Rajendran, G., Smith, T., Lemon, O., Porayska-Pomsta, K., ... & Bernardini, S. (2011, June). Social communication between virtual characters and children with autism. In international conference on artificial intelligence in education (pp. 7-14). Springer, Berlin, Heidelberg.

  1. Thank you for your comment and suggestion about matching factors. In the revised manuscript, we considered them in the limitations section of the Discussion, as suggested by you. In future work, we consider these factors as moderators in the models.
